# Joint Use of PROSAIL and DART for Fast LUT Building: Application to Gap Fraction and Leaf Biochemistry Estimations over Sparse Oak Stands

**Thomas Miraglio [1,2,\*], Karine Adeline [1] , Margarita Huesca [3], Susan Ustin [3] and Xavier Briottet [1]**

[1] Office National d'Études et de Recherches Aérospatiales (ONERA), 2 Avenue Edouard Belin, 31055 Toulouse, France; karine.adeline@onera.fr (K.A.); Xavier.Briottet@onera.fr (X.B.)

[2] Université Fédérale Toulouse Midi-Pyrénées, 41 Allées Jules Guesde, 31013 Toulouse, France

[3] John Muir Institute of the Environment, University of California, Davis, One Shield Avenue, Davis, CA 95616, USA; mhuescamartinez@ucdavis.edu (M.H.); slustin@ucdavis.edu (S.U.)

[\*] Correspondence: thomas.miraglio@onera.fr

**Abstract:** Gap Fraction, leaf pigment contents (content of chlorophylls *a* and *b* ($C_{ab}$) and carotenoids content (Car)), Leaf Mass per Area (LMA), and Equivalent Water Thickness (EWT) are considered relevant indicators of forests' health status, influencing many biological and physical processes. Various methods exist to estimate these variables, often relying on the extensive use of Radiation Transfer Models (RTMs). While 3D RTMs are more realistic to model open canopies, their complexity leads to important computation times that limit the number of simulations that can be considered; 1D RTMs, although less realistic, are also less computationally expensive. We investigated the possibility to approximate the outputs of a 3D RTM (DART) from a 1D RTM (PROSAIL) to generate in very short time numerous extensive Look-Up Tables (LUTs). The intrinsic error of the approximation model was evaluated through comparison with DART reference values. The model was then used to generate LUTs used to estimate Gap Fraction, $C_{ab}$, Car, EWT, and LMA of Blue Oak-dominant stands in a woodland savanna from AVIRIS-C data. Performances of the approximation model for estimation purposes compared to DART were evaluated using Wilmott's index of agreement ($d_r$), and estimation accuracy was measured with coefficients of determination ($R^2$) and Root Mean Squared Error (RMSE). The low approximation error of the proposed model demonstrated that the model could be considered for canopy covers as low as 30%. Gap Fraction estimations presented similar performances with either DART or the approximation ($d_r$ 0.78 and 0.77, respectively), while $C_{ab}$ and Car showed improved performances ($d_r$ increasing from 0.65 to 0.77 and 0.34 to 0.65, respectively). No satisfying estimation methods were found for LMA and EWT using either models, probably due to the high sensitivity of the scene's reflectance to Gap Fraction and soil modeling at such low LAI. Overall, estimations using the approximated reflectances presented either similar or improved accuracy. Our findings show that it is possible to approximate DART reflectances from PROSAIL using a minimal number of DART outputs for calibration purposes, drastically reducing computation times to generate reflectance databases: 300,000 entries could be generated in 1.5 h, compared to the 12,666 total CPU hours necessary to generate the 21,840 calibration entries with DART.

**Keywords:** leaf mass per area; equivalent water thickness; chlorophylls; carotenoids; open canopy

## 1. Introduction

Several structural and biochemical parameters of plants, such as Gap Fraction, leaf chlorophylls a+b and carotenoids contents ($C_{ab}$ and Car), leaf mass per area (LMA), or leaf equivalent water

thickness (EWT), are recognized indicators of plants' health status [1–4]. They influence many biological and physical processes such as photosynthetic activity, nutrient cycles, gross primary production, rainfall interception, and heat fluxes [5–7].

While field and laboratory measurements can only provide limited information on these indicators in both time and space, multispectral and hyperspectral remote sensing data have been extensively used to estimate canopy structural and biochemical parameters over large areas and can allow for recurrent measurements of the study sites [8–10]. Hyperspectral sensors measure forest canopy reflectance using numerous spectral bands over the solar spectrum, so that slight variations of the reflected radiation can be detected. Common estimation methods of Gap Fraction and leaf biochemical properties belong to three main families: empirical-statistical methods calibrate a model to validation data acquired in-field, physical approaches rely on the inversion of radiation transfer models (RTMs) that simulate canopy reflectance, and hybrid methods bring together the fine-tuning of physically-based approaches with the flexibility of empirical-statistical methods. Baret and Buis [11] and Verrelst et al. [12] provide reviews of the various estimation methods, their respective difficulties, and the current solutions to try to overcome them.

In many cases, data are insufficiently available to calibrate an empirical-statistical method, and it is necessary to turn to physical or hybrid approaches and rely on RTMs. This can prove to be computationally demanding, as a consequent number of simulations could be necessary so that acceptable accuracy in variable retrieval is achieved. An important number of RTMs, either using homogeneous or heterogeneous scenes (hereafter designated as "1D" and "3D" RTMs), are available and several took part in the RAdiation transfer Model Intercomparison (RAMI) experiments [13–16]. 1D RTMs are adapted to homogeneous scenes and are by design limited in the number of possible variable parameters. While not very realistic, this makes for very short computation time and overall easier inversions for ecosystems with medium to high canopy covers. On the contrary, 3D RTMs provide a detailed description of the canopy layers and components through many variables that can be either fixed by the user (using a priori knowledge) or kept as variable parameters. This is of prime importance in particular for the modeling of sparse forests and tree–grass ecosystems that are widely distributed on Earth [17], as the spectral contribution of the canopy to the total scene reflectance is limited and ground and shadows are more visible to the sensors. However, they most commonly rely on ray tracing methods and this added complexity can lead to a dramatic increase in computation times, which limits the sampling schemes that can realistically be considered for each variable of interest.

Unfortunately, no single best sampling scheme has been identified so far: Ali et al. [18] considered a uniform distribution of the variables over their respective ground-truth ranges when working with INFORM [19]; Weiss et al. [20] drew each parameter's values according to a distribution law which was proportional to the reflectance's sensitivity to the parameter; Ali et al. [21] used multivariate normal distributions and covariance matrices produced from ground truth data with INFORM; Hernández-Clemente et al. [22] used both monovariate and multivariate random samplings with DART [23]. Due to the computation times of 3D RTMs, testing multiple sampling schemes when building Look-Up Tables (LUTs) with tens of thousands of entries is not realistically feasible. Being able to consistently optimize the LUT sampling scheme at low time cost, either for direct LUT-based inversion or subsequent training of a machine-learning model, could therefore prove beneficial.

The aim of this study was to combine the realism of 3D RTM with the speed of 1D RTM to be able to quickly generate LUTs with several variable parameters and arbitrary sampling. To do so, the PROSAIL [24] (1D) and DART (3D) canopy RTMs were considered. Both PROSAIL and DART were used to calibrate a model that approximates DART reflectance outputs from PROSAIL's. The performances of this model (named PROSAIL2DART) were assessed by comparing its outputs with DART reference values. PROSAIL2DART was then used to estimate Gap Fraction, oak leaf pigment content, and oak LMA and leaf EWT over a woodland savanna. Estimations accuracies were assessed by confronting estimations with field measurements done at various stands and dates.

## 2. Materials and Methods

### 2.1. Study Site

The study site is an oak woodland savanna located in the lower foothills of the Sierra Nevada Mountains (Tonzi Ranch, latitude: 38.431°N; longitude: 120.966°W; altitude: 177 m; average slope: 1.5°). It has a Mediterranean climate alternating between mild, wet winters and hot, dry summers. Ninety percent of the overstory are Blue Oak (*Quercus douglasii*—QUDO), the remaining 10% being mostly Grey Pine (*Pinus sabiniana*—PISA). Blue oaks are deciduous, their leaves start to sprout in April and have been shed by November. The mean canopy cover (CC) of the site is 47%, and the mean LAI is 0.8. The understory is composed of cool season annual C3 grass species active from December to May and dry during summer and autumn. Both oak trees and grasses are active in April and May. The soil is an Auburn very rocky silt loam (Lithic haploxerepts). More detailed site information can be found in previous studies [25–27].

As PISA only represent 10% of the overstory, the present study Gap Fraction plots contained either only QUDO or a QUDO-PISA mix with a QUDO majority (Section 2.2.1). Leaf collection for EWT, LMA and leaf pigment content estimation were done in a pure-QUDO part of the site (Section 2.2.2). Only QUDO were modeled in the RTM, as correctly modeling coniferous trees in 3D RTMs can be challenging [28] and no PISA-dominant plots were included in the study (three stands are mixed and PISA canopy cover represents only 37% of the total canopy cover in the most mixed stand).

Figure 1 shows an aerial view of the site and the locations of the various plots used in this study. Picture of both QUDO and PISA as well as average dimensions of QUDO are shown in Figure 2a,b.

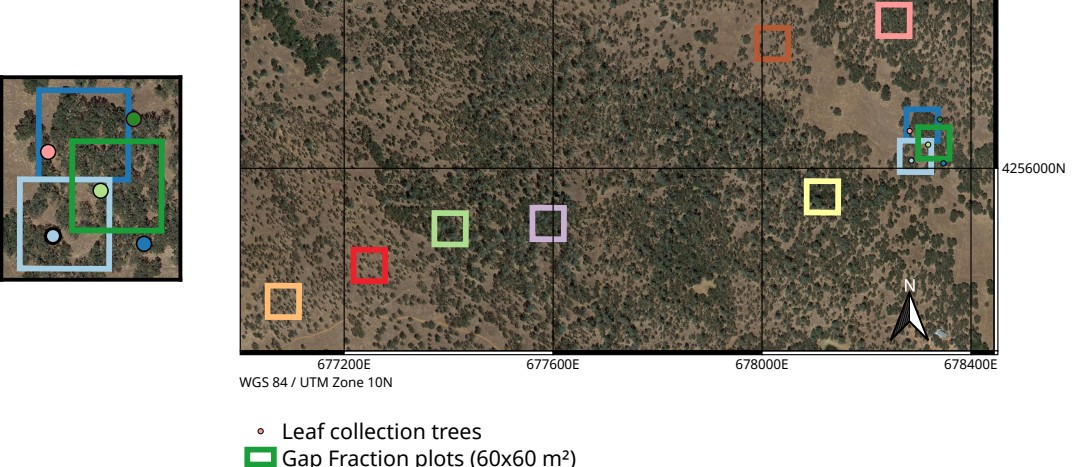

**Figure 1.** Aerial view (**right**) of the study site and zoom-in on the leaf collection trees (**left**). Different Gap Fraction plots and leaf collection trees are identified with different colors.

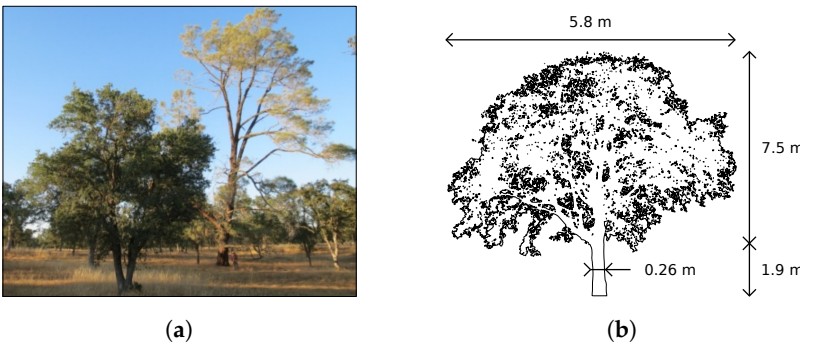

(**a**)　　　　　　　　　　　　　　　　　　　　　　　　　(**b**)

**Figure 2.** Panel (**a**), a picture of both a Blue Oak (foreground) and a Grey Pine (background). Panel (**b**), average dimensions of Blue Oaks at Tonzi Ranch.

*2.2. Field Data*

2.2.1. Gap Fraction

Field data were collected coincident with the NASA Hyperspectral Infrared Imager (HyspIRI) Mission Study Airborne Campaigns that took place in September 2013 and June 2014 and 2016 (https://hyspiri.jpl.nasa.gov/). Several digital hemispherical photographs (DHP) were collected over multiple 60 m × 60 m plots across the study site covering all vegetation cover fractions and species composition. These plots were selected to span the full variation in species composition and canopy density. Information concerning the number of plots for each date is given in Table 1.

From each 60 m × 60 m plot, nine DHP were taken using a Nikon Coolpix 4300 camera post sunset when no direct sunlight was visible. The DHP were taken according to the sampling patterns shown in Figure 3. DHP were processed using CAN-EYE (https://www6.paca.inrae.fr/can-eye). CAN-EYE calculated the Gap Fraction with azimutal and zenithal resolutions of 2.5°, using a circle of interest of 65°. The theory behind CAN-EYE estimations is described by Weiss et al. [29] and is also available at https://www6.paca.inrae.fr/can-eye/Documentation/Documentation. As the LAI of the site is very low, there is no significant risk of Gap Fraction saturation as this phenomenon happens for LAI higher than 5.

$$\text{EWT} = \frac{\text{fresh weight} - \text{dry weight}}{\text{leaf area}} \tag{1}$$

$$\text{LMA} = \frac{\text{dry weight}}{\text{leaf area}} \tag{2}$$

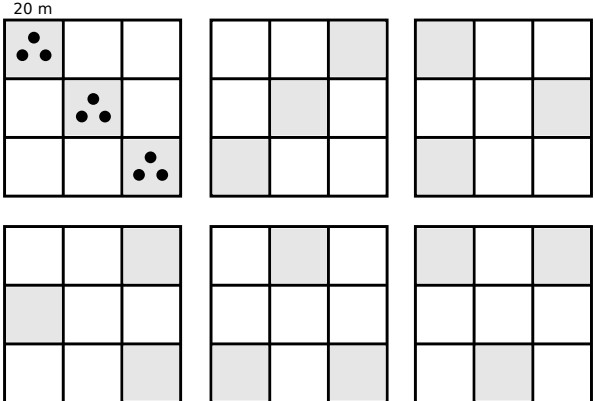

**Figure 3.** Possible sampling patterns of the Gap Fraction plots. For each plot, three digital hemispherical photographs (DHP) (black dots) were taken in a North–South East–Southwest pattern 10 m apart in each of the three randomly selected 20 m × 20 m subplots (shaded in gray).

**Table 1.** Field data used in this study for Gap Fraction and leaf biochemistry and associated dates of collection. In bold, data used for validation in the present study.

|  | Field Data | |
|---|---|---|
| **Date** | **Gap Fraction Plots** | **Biochemistry Trees** |
| June 2013 |  | **3** |
| September 2013 | **2** | **5** |
| April 2014 |  | 5 |
| June 2014 | **8** | **5** |
| October 2014 |  | 5 |
| June 2016 | **7** |  |
| Validation data | 17 | 13 |

### 2.2.2. Equivalent Water Thickness, Leaf Mass Per Area, and Leaf Biochemistry Measurements

To retrieve EWT, LMA, and leaf biochemistry, a set of fully expanded leaves was collected five healthy QUDO individuals presenting a structure typical of the site. Leaves were collected from the upper, sunlit portion of the canopy. Sampling started within an hour of the timing of the overflight. Leaf samples were collected from open grown trees that were in full sunlight, as high into the canopy as possible, and from branches on the east and west sides of the tree. Attention was paid to ensure that collected leaves were healthy, and collection always occurred during dry days. Leaves were placed in a plastic bag and stored on blue ice (or in a lab refrigerator) until lab measurements could be made (<48 h). Plastic bags were weighed with a mg precision scale before going to the field. In the lab, the bags with leaves inside were weighed, with the weight difference giving leaf fresh weight. After that, the thickness of each leaf was measured using a caliper and all leaves were scanned in TIF format with 150 dpi. Leaf area was estimated using the scanned image in TOASTER software. Finally, all the leaves were put into a paper bag to dry at 65 degrees Celsius until the weight did not change when the leaves were reweighed (two to three days) to obtain leaf dry weight. Finally, EWT and LMA were calculated according to Equations (1) and (2).

The methodology for leaf $C_{ab}$ and Car retrieval has been described by Miraglio et al. [30].

### 2.2.3. Trunk Reflectances

Tree trunk reflectances were collected from the five individuals from the leaf biochemistry collection and measured with an Analytical Spectral Device (ASD; ASD Inc., Boulder, CO, USA) contact probe. A spectralon panel was used for calibration purposes before every acquisition. Trunk reflectances were obtained over the 0.350 to 2.500 μm spectral range. Small portions of the trunk were collected and situated in a horizontal surface to facilitate the measurement

### 2.2.4. Airborne Hyperspectral Remote Sensing Data

AVIRIS-C hyperspectral data are processed and delivered by NASA Jet Propulsion Laboratory (JPL; http://aviris.jpl.nasa.gov). Table 2 gives information about the date of the acquisitions used in this study. Images were acquired at nadir 20 km above the ground within $\pm 1$ h of the solar noon to avoid spectral directional effects. Preprocessing steps provided by NASA JPL included radiometric calibration, geometrical orthorectification, nearest neighbor spatial resampling at 18 m, and atmospherical correction performed with ATREM [31], in order to retrieve surface reflectance. The AVIRIS-C images used in this study were co-registered and spectral temporal corrections were applied using the same protocol as in Miraglio et al. [30]. Hyperspectral images from April and October 2014 were also available but could not be used in this study, as grass was still green in April, which would have led to additional complexity when doing Gap Fraction and leaf biochemistry estimations, and a fire plume was above the site in October at the time of the airborne acquisitions.

**Table 2.** Description of AVIRIS-C airborne acquisitions.

| Year | Date (DOY [1]) | Time (PDT [2]) |
|---|---|---|
| 2013 | 4 June (155) | 12h30 p.m. |
|  | 19 September (262) | 12h40 p.m. |
| 2014 | 2 June (153) | 12h00 p.m. |
| 2016 | 9 June (161) | 12h30 p.m. |

[1] Day of Year; [2] Pacific Daylight Time.

### 2.3. PROSAIL2DART

#### 2.3.1. Methodology

Both PROSAIL and DART were used to compute scene reflectances over a calibration (LAI, $C_{ab}$, Car, LMA, EWT) grid hereafter designated as *cal*. Band reflectance ratios $\frac{\rho_{DART}}{\rho_{PROSAIL}}$ were computed for each point of the grid and used to calibrate linear 5-D interpolators. These interpolators are used to transform any reflectance obtained with PROSAIL into a reflectance similar to what DART would have obtained. For each date, one 5-D interpolator was computed for each (CC, understory reflectance, Anthocyanins content ($C_{ant}$)) triplet as these parameters were either not possible to model in PROSAIL (CC) or to limit the uncertainties (understory reflectance, $C_{ant}$). This calibration/transformation method is hereafter called PROSAIL2DART (P2D). A diagram of the methodology is shown in Figure 4.

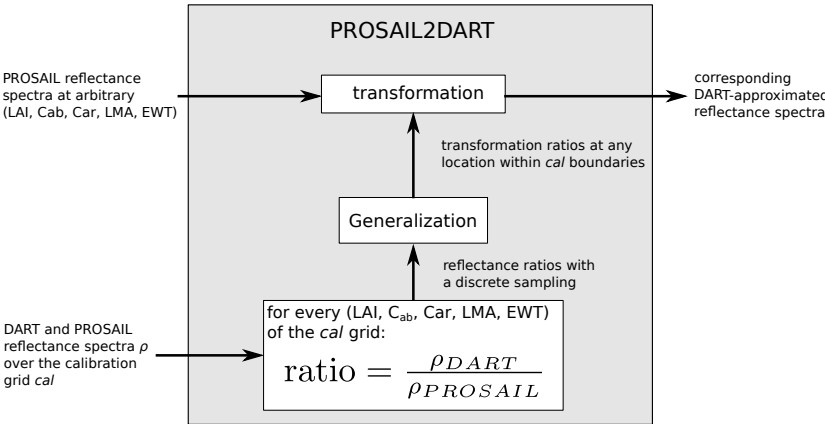

**Figure 4.** PROSAIL2DART methodology.

#### 2.3.2. RTM Parametrization

**DART**

DART version 5.7.3v1078 was used to simulate canopy reflectances. DART is a radiation transfer model able to simulate light interactions and multiple scattering effects within a 3D scene, including the topography and the atmosphere. DART includes PROSPECT to model leaf reflectance and transmittance. A precise description of the DART model can be found in Gastellu-Etchegorry et al. [32] and Gastellu-Etchegorry et al. [23]. Trees are defined by structural parameters such as the shape and size of their crown or the distribution and optical properties of their leaves.

The scene modeling done in this study is based on the same simplified forest representation as the one done in Miraglio et al. [30]: canopy is represented with 4 lollipop trees and the ground is modeled as a lambertian surface, the reflectance of which was extracted from AVIRIS-C images, as this previously proved sufficient to estimate both LAI and leaf pigment content at the AVIRIS-C spatial resolution. Different soil reflectance were considered: from the sets of pure soil pixels extracted from open parts of the site, mean and mean ± standard deviation reflectances were used to build the LUTs in order to better take into account possible ground reflectance variations over the site. For simplicity purposes and to ensure that (i) the (LAI, $C_{ab}$, Car, LMA, EWT) space that could be covered by P2D was an hyperrectangle and (ii) the density of calibration samples was uniform over this space, *cal* followed a regular sampling scheme. Tables 3 and 4 describe the various inputs used to create the DART scenes.

The Gap Fractions of the DART scenes were computed for all combinations of CC and LAI. It was retrieved using the DART 3D Radiative Budget tool, by considering the percentage of diffuse illumination intercepted by the ground. The specific DART parameters to obtain these results are

- illumination using a single wavelength,
- no radiative transfer in the atmosphere,
- SKYL (atmospheric scattering of sun radiance) set to 1,
- number of iterations set to 0, and
- smaller mesh size of irradiance sources set to 0.005 m

Gap Fraction can be considered a function of CC an LAI, i.e., Gap Fraction $= f(\text{CC}, \text{LAI})$. Therefore, when generating the P2D, fine LUTs Gap Fractions were derived from the (CC, LAI) values by linear interpolation, using the *test* and *cal* values as reference.

**Table 3.** DART-fixed scene parameters used in this study for the calibration and test Look-Up Tables (LUTs). PROSAIL used the same LAD and sun zenith/azimuth as DART.

| Parameter | Value |
|---|---|
| Voxel size x, y, z (m) | 0.4, 0.4, 0.4 |
| Tree height (m) | 9.4 |
| Crown shape | ellipsoidal |
| Crown diameter (m) | 5.8 |
| Crown height (m) | 7.5 |
| Trunk height (m) | 6.63 |
| Trunk dbh (m) | 0.26 |
| LAD | spherical |
| Sun zenith/azimut | according to acquisition date |

**Table 4.** DART, PROSAIL, and PROSPECT variable parameters used for the calibration (*cal*) and test (*test*) LUTs. For $C_{ant}$, values between parentheses concern the September 2013 LUT, other values concern all June LUTs.

| Parameter | Values/Range | | Step | |
|---|---|---|---|---|
| | *cal* | *test* | *cal* | *test* |
| CC (%) | 10–90 | 10–90 | 20 | |
| LAI (m$^2$/m$^2$) | 0.1–1.9 | 0.25–1.75 | 0.3 | |
| $C_{ab}$ (µg/cm$^2$) | 10–60 | 15–55 | 10 | |
| Car (µg/cm$^2$) | 2–14 | 4–12 | 4 | |
| LMA (g/cm$^2$) | 7–16 $\times$ 10$^{-3}$ | 8.5–14.5 $\times$ 10$^{-3}$ | 3$\times$ 10$^{-3}$ | |
| EWT (cm) | 5–17 $\times$ 10$^{-3}$ | 7–15 $\times$ 10$^{-3}$ | 4 $\times$ 10$^{-3}$ | |
| $C_{ant}$ (µg/cm$^2$) | 0(0–2) | 0 | 0(2) | 0 |
| Ground Reflec. | mean, mean $\pm$ Std | June 2016 mean | | |

PROSAIL

The 4SAIL version of PROSAIL was used in the present study, using a Python wrapper (https://github.com/jgomezdans/prosail, DOI: 10.5281/zenodo.2574925). PROSAIL combines the leaf model PROSPECT with the 1-D turbid canopy RTM SAIL. A thorough description and history of the PROSAIL model can be found in Berger et al. [33]. Leaf angle distribution (LAD) is not a direct input of PROSAIL and both the average leaf slope LIDFa and the associated distribution bimodality LIDFb must be given. A spherical LAD is obtained with LIDFa = −0.35 and LIDFb = −0.15. Ground reflectance was the same reflectance as that given to DART, as were the solar zenith and azimuth angles. LAI variations were the same as those given to DART.

PROSPECT

Leaf optical properties were simulated using the PROSPECT model, which is implemented in DART and PROSAIL. PROSPECT-D [34] was used in this study, and a small $C_{ant}$ was introduced as a possible case for September to take into account possible leaf senescence. The leaf structure parameter N was set to 1.8. The PROSPECT specific parameters considered in this study are given in Table 4.

### 2.3.3. Error Assessment

As the P2D linear interpolators were calibrated on a regular grid, the maximum differences between P2D and DART reflectances are located at the centers of the hypercubes defined by the *cal* grid. The P2D approach will be validated if the difference between P2D and DART reflectances at the hypercube center are negligible.

Let *test* designate the set of the hypercubes centers, which correspond to the *test* grid. Let $P2D^{test}$ and $D^{test}$ be the reflectances computed by P2D and DART on *test* points and $D^{cal}$ the reflectances computed by DART on *cal* points, which correspond to the hypercubes corners. The P2D approximation was evaluated using the *E* ratio, computed as

$$E = \underset{i}{\text{median}} \frac{\left| P2D_i^{test} - D_i^{test} \right|}{\min_{j} \left| D_i^{test} - D_{i,j}^{cal} \right|_{>0.001}} \times 100 \tag{3}$$

with *i* the hypercube identifier and *j* the hypercube corner identifier. This ratio was designed to compare the reflectance distance between P2D and DART at the hypercube center (maximum error) with the reflectance distance between the hypercube center and corners obtained through DART. A value close to 100 indicates that the P2D error is similar to the smallest difference between the hypercube's center and corners, while a value close to 0 indicates that the P2D error is negligible. An illustration of the P2D validation methodology is presented Figure 5. An *E* value lower than 50 indicates that the P2D approximation is closer to the hypercube center than the corners. A condition that $\left| D_i^{test} - D_{i,j}^{cal} \right|$ should be non-negligible (>0.001 when reflectance range is 0 to 1) was used as variables do not necessarily have an influence at all wavelengths, which could lead to close to zero differences between some corners and the center and make *E* diverge erroneously.

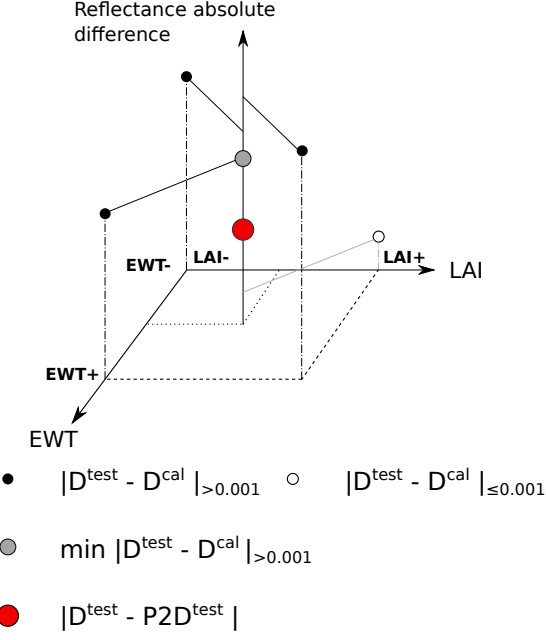

**Figure 5.** Illustration of the P2D validation methodology on a 2D grid, with calculation of the various differences necessary to compute *E* for one hypercube.

### 2.4. Fine Lut Building

P2D was subsequently used to generate a fine DART-like LUT for each AVIRIS-C image, following a Latin Hypercube sampling. The correlation between $C_{ab}$ and Car visible in the field data was taken

into account by constraining the Car values around 2.5 times the standard deviation between field data and the regression line (see Figure 6). For LAI, $C_{ab}$, LMA and EWT are the boundaries of the Latin Hypercube corresponding to those of the calibration LUTs presented in Table 4. Only CC 30 to 90% were considered, and for every ground reflectance and $C_{ant}$ value, 50,000 cases were generated and distributed equally among the CC. Therefore, June 2013, 2014, and 2016 fine LUTs each have 150,000 entries, and September 2013 LUT has 300,000 entries.

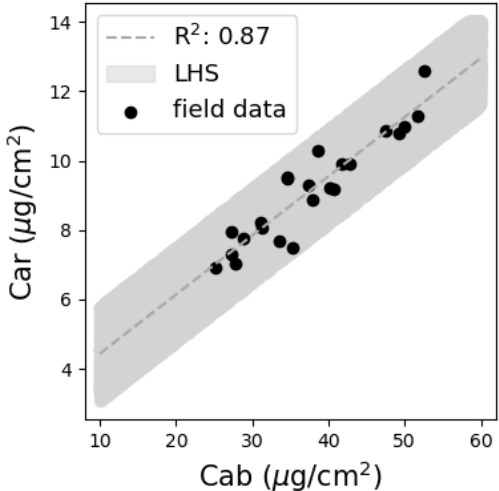

**Figure 6.** Relationship between Car and $C_{ab}$ in the field data (dots) and as implemented (shaded area) with the Latin Hypercube sampling.

### 2.5. Lut-Based Inversions

To assess the performance improvement when using P2D instead of simply using DART with a regular sampling scheme, both DART *cal* and P2D fine LUTs were used to retrieve Gap Fraction and leaf biochemistry. LUT-based approaches consist in finding the simulated reflectance $\hat{y}$ that is the most similar to the measured one, $y$, according to a cost function. Several cost functions were selected for this study: root mean square error (RMSE; Equation (4)), spectral angle mapper (SAM; Equation (5)), and vegetation index (VI) differences ($D_{VI}$; Equation (6)).

$$RMSE(y, \hat{y}) = \sqrt{\frac{1}{N}\Sigma_{i=1}^{N}(y_i - \hat{y}_i)^2} \tag{4}$$

$$SAM(y, \hat{y}) = cos^{-1}\left(\frac{\Sigma_{i=1}^{N}y_i\hat{y}_i}{\sqrt{\Sigma_{i=1}^{N}\hat{y}_i\hat{y}_i}\sqrt{\Sigma_{i=1}^{N}y_iy_i}}\right) \tag{5}$$

$$D_{VI}(y, \hat{y}) = abs\left(VI(y) - VI(\hat{y})\right) \tag{6}$$

RMSE and SAM were computed using variable-specific spectral intervals. The Gap Fraction interval covered the near-infrared (NIR) and short wavelength infrared (SWIR) (INT GAP, 0.8–2.45 µm). Intervals for $C_{ab}$ and Car were parts of the visible range (INT CAB, 0.5–0.75; INT CAR, 0.5–0.55 µm), while those used for LMA and EWT were parts of the NIR and SWIR (INT LMA, 0.8–1.3; INT EWT, 1.3–2.45 µm). The spectral intervals were chosen based on their sensitivity to the variables of interest according to the results of a Sobol sensitivity analysis on the DART *cal* LUTs (not shown).

Before LUT-based inversion, VI capabilities to estimate the variables of interest were assessed by fitting a function between VI and variables ($VI = v^a + b$ for biochemistry and $VI = a \times v + b$ for Gap Fraction, with $v$ the variable's value). If no relationship could be found, the VI was not considered for the inversion.

Table 5 shows all the cost functions tested for this study for each variable of interest, as well as the goodness of the fit of each VI when applicable. Estimation results were computed as the mean of multiple best solutions. The number of solutions considered for each LUT was 0.5% of the LUT size.

**Table 5.** Cost functions used in this study, best $R^2$ obtained across all LUTs for the vegetation index (VI) and associated performances for the LUT-based (DART and P2D) inversions (when applicable).

| | VI Source | VI $R^2$ | $d_r$ DART *cal* | $d_r$ P2D fine LUT |
|---|---|---|---|---|
| **Gap fraction** | | | | |
| RMSE INT GAP | | | 0.75 | 0.75 |
| SAM INT GAP | | | 0.76 | 0.76 |
| $D_{NDVI}$ | Tucker [35] | 0.96 | 0.78 | 0.77 |
| $D_{MSAVI2}$ | Qi et al. [36] | 0.9 | 0.76 | 0.74 |
| **$C_{ab}$** | | | | |
| RMSE INT CAB | | | 0.49 | 0.31 |
| SAM INT CAB | | | 0.12 | 0.23 |
| $D_{TCARI/OSAVI}$ | Haboudane et al. [37] | 0.81 | 0.25 | 0.23 |
| $D_{Maccioni}$ | Maccioni et al. [38] | 0.91 | 0.57 | 0.69 |
| $D_{gNDVI}$ | Smith et al. [39] | 0.76 | 0.65 | 0.75 |
| $D_{GM\_94b}$ | Gitelson and Merzlyak [40] | 0.6 | 0.65 | 0.77 |
| $D_{MCARI2}$ | Haboudane et al. [41] | 0.01 | | |
| **Car** | | | | |
| RMSE INT CAR | | | −0.08 | 0.2 |
| SAM INT CAR | | | −0.37 | 0.53 |
| $D_{R515\_R570}$ | Hernández-Clemente et al. [22] | 0.29 | | |
| $D_{CRI550}$ | Gitelson et al. [42] | 0.09 | | |
| $D_{TCARI/OSAVI}$ | Haboudane et al. [37] | 0.72 | 0.04 | 0.27 |
| $D_{Maccioni}$ | Maccioni et al. [38] | 0.81 | 0.12 | 0.59 |
| $D_{gNDVI}$ | Smith et al. [39] | 0.68 | 0.11 | 0.64 |
| $D_{GM\_94b}$ | Gitelson and Merzlyak [40] | 0.61 | 0.34 | 0.65 |
| $D_{MCARI2}$ | Haboudane et al. [41] | 0.01 | | |
| **LMA** | | | | |
| RMSE INT LMA | | | 0.03 | 0.19 |
| SAM INT LMA | | | 0.29 | 0.24 |
| $D_{lma\_ND}$ | le Maire et al. [43] | 0 | | |
| $D_{lma\_D}$ | le Maire et al. [43] | 0.5 | −0.34 | 0.1 |
| $D_{NDNI}$ | Serrano et al. [44] | 0 | | |
| $D_{NDLI}$ | Serrano et al. [44] | 0 | | |
| **EWT** | | | | |
| RMSE INT EWT | | | −0.32 | 0.04 |
| SAM INT EWT | | | −0.49 | 0.29 |
| $D_{EVI}$ | Huete et al. [45] | 0 | | |
| $D_{NDWI}$ | Gao [46] | 0.01 | | |
| $D_{SIWSI}$ | Fensholt and Sandholt [47] | 0.02 | | |
| $D_{NDI7}$ | Trombetti et al. [48] | 0.01 | | |
| $D_{NDII}$ | Hardisky et al. [49] | 0 | | |
| $D_{SRWI}$ | Zarco-Tejada et al. [50] | 0.01 | | |
| $D_{MSI}$ | Hunt and Rock [51] | 0 | | |
| $D_{SWIRR}$ | Trombetti et al. [48] | 0.01 | | |
| $D_{WI}$ | Penuelas et al. [52] | 0.01 | | |

## 2.6. Validation Metrics

Gap fraction, leaf pigments content, EWT, and LMA estimates were compared with the field measurements available using the following criteria; total RMSE, systematic and unsystematic RMSE (Willmott [53]), the model performance index $d_r$ (Willmott et al. [54]), and $R^2$ of the predicted vs. measured regression line.

Concerning Gap Fraction, each validation point was the average of Gap Fraction values derived from hemispherical pictures taken within a 60 m × 60 m plot (see Section 2.2.1). Direct comparison between pixel-estimated value and validation data can be inappropriate as the area covered by the DHP of a plot is wider than the AVIRIS-C pixel (18 m × 18 m). Therefore, validation values were compared to the average of the Gap Fractions estimated over a 3 × 3 pixels windows centered on the pixel corresponding to the plot center, similarly to the method employed in Miraglio et al. [30].

Biochemistry validation data were obtained at the leaf scale, for one tree in each validation pixel. It was assumed that biochemistry estimations could be directly extracted from the pixels associated with the acquisition positions.

## 3. Results

### 3.1. Comparison between Aviris-C and Dart Reflectances

Figure 7 shows the validation pixels' reflectances and compares them to the reflectance extrema found in their corresponding LUT. For June 2013, September 2013 and June 2016, all pure pixel reflectances fall within the extrema of the LUTs whatever the wavelength. One reflectance from a mixed plot is severely out of the boundaries of the LUT, while up to 8 other mixed plot reflectances are slightly below the LUT minima between 0.9 and 1.6 µm, for a total of at most 9 reflectances below the LUT minima at some wavelengths out of 63 for June 2016. For June 2014, almost all reflectances including those from mixed pixels also fall within the extrema: of the 68 pixel reflectances, 6 are below the LUT minima around 1.24 µm and 1.6 µm, with a maximum difference of 0.02.

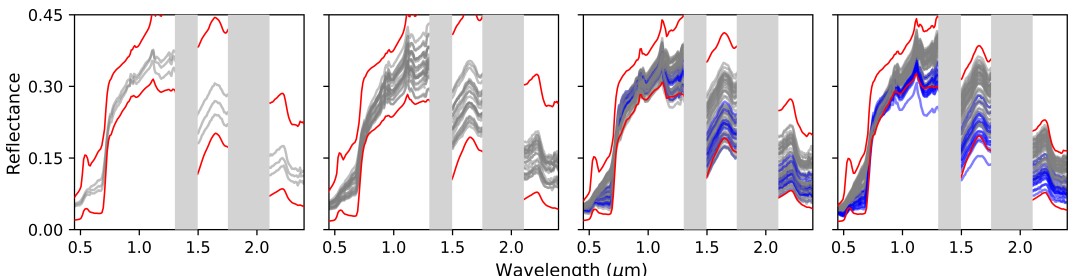

**Figure 7.** LUT reflectances and AVIRIS-C reflectances at validation pixels for each date (from left to right: June 2013, September 2013, June 2014, June 2016). In red, reflectance boundaries of the associated LUT, in gray, AVIRIS-C reflectances. In blue, AVIRIS-C reflectances from mixed QUDO-PISA validation pixels.

### 3.2. PROSAIL2DART Errors

Figure 8 and Table 6 show the evolution of the $E$ ratio over the CC and wavelengths. In the visible, all wavelengths are well approximated by P2D for CC $\geq$ 30%, with the highest $E$ value being 21% at 0.68 µm and 30% CC. While for 10% CC, the green and NIR are also well approximated ($E < 50\%$), this is not the case for the blue and red regions where $E$ can be above 50%. In the SWIR, for 10% CC $E$ values are considerably below 50% for $\lambda < 1.8$ µm. However, higher values are found at higher wavelengths and the maximum, 46%, is obtained at 1.49 µm. Estimations with either DART *cal* or P2D LUTs only used the CC $\geq$ 30% cases to avoid uncertainties.

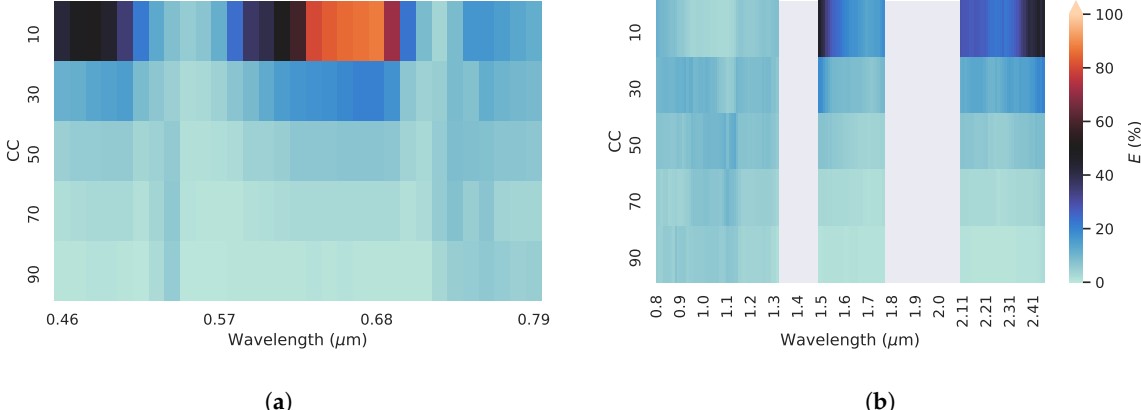

**(a)** **(b)**

**Figure 8.** P2D error for each wavelength and CC over the (**a**) visible and (**b**) near-infrared (NIR) spectral regions. Panels (**a**,**b**) share the same color bar.

**Table 6.** Maximum value of the $E$ ratio and associated wavelength for each CC over the visible (VIS) and NIR spectral ranges.

| CC | Maximum $E$ (%) | | Wavelength (μm) | |
|---|---|---|---|---|
| | **VIS** | **NIR** | **VIS** | **NIR** |
| 10 | 87 | 46 | 0.68 | 1.49 |
| 30 | 21 | 21 | 0.68 | 2.45 |
| 50 | 9 | 13 | 0.73 | 1.13 |
| 70 | 8 | 10 | 0.73 | 1.10 |
| 90 | 6 | 8 | 0.76 | 0.88 |

### 3.3. PROSAIL2DART Fine Lut Generation

It took 12,666 h (total CPU time of a server equipped with Broadwell Intel® Xeon® CPU E5-2650 v4 @ 2.20 GHz) to generate the 21,840 reflectances required to build the DART *cal* LUT dedicated to September 2013 (the most extensive LUT, as two values of anthocyanins are also considered). For comparison, once P2D was calibrated (the calibration time is negligible), it took 1.5 h (total CPU time on a computer equipped with an Intel® Core™ i5-6300HQ CPU @ 2.30 GHz) to generate the 300,000 entries of the P2D fine LUT.

### 3.4. Estimation Performances

Table 5 shows the best $R^2$ achieved by the various VI when fitted over each LUT. Gap Fraction was very well measured by its VI, and overall so were C$_{ab}$ and Car, with only MCARI2, R515_570, and CRI550 presenting no relation with the estimates' values. Concerning LMA and EWT, no satisfying relation could be found, and only lma_D could find a slight relation with a $R^2$ of 0.5. Only the inversion methods with $R^2$ higher than 0.5 were considered suitable candidates for LUT-based inversions.

Concerning Gap Fraction, both LUTs perform in equivalent manner, with good performances whatever the method (maximum $d_r$ is 0.78 for $D_{NDVI}$ applied on DART *cal* and 0.77 when applied on the P2D LUT). C$_{ab}$ estimations show improved performances with the P2D LUTs for all cost functions except RMSE INT CAB and $D_{TCARI/OSAVI}$, which have slightly lower $d_r$. $D_{GM\_94b}$ offers the best performances, with $d_r = 0.77$ when applied on the P2D LUTs. For Car, the best $d_r$ is also obtained with the P2D LUTs with $D_{GM\_94b}$, and P2D consistently improved the $d_r$. Concerning LMA and EWT, the P2D fine LUTs appear to slightly improve performances for the selected cost functions (at the exception of SAM INT LMA that decreases slightly); however, $d_r$ remains low. Their respective best-performing cost functions are SAM INT LMA with DART *cal* and SAM INT EWT with P2D.

### 3.5. Estimation Plots

Figure 9 compares estimated and field values for the various estimates of this study when using the best performing methods identified in Section 3.4. Gap Fraction estimations present both a high $R^2$ and a low RMSE (0.78 and 0.1, respectively). While it appears that one of the June 2014 mixed plots (yellow) was overestimated, other mixed plots present a similar behavior as pure QUDO plots. Similar behaviors are found for $C_{ab}$ and Car: most of the points seem to follow the first bisector, and the point with the highest $C_{ab}$ and Car values is slightly underestimated. RMSE errors in both cases remain low (4.14 and 1.05 µg/cm$^2$, respectively). No trend between estimations and field data could be found for either LMA and EWT (their respective $R^2$ are 0.14 and 0.01) and RMSE of $2.2 \times 10^{-3}$ g/cm$^2$ and $2 \times 10^{-3}$ cm are obtained.

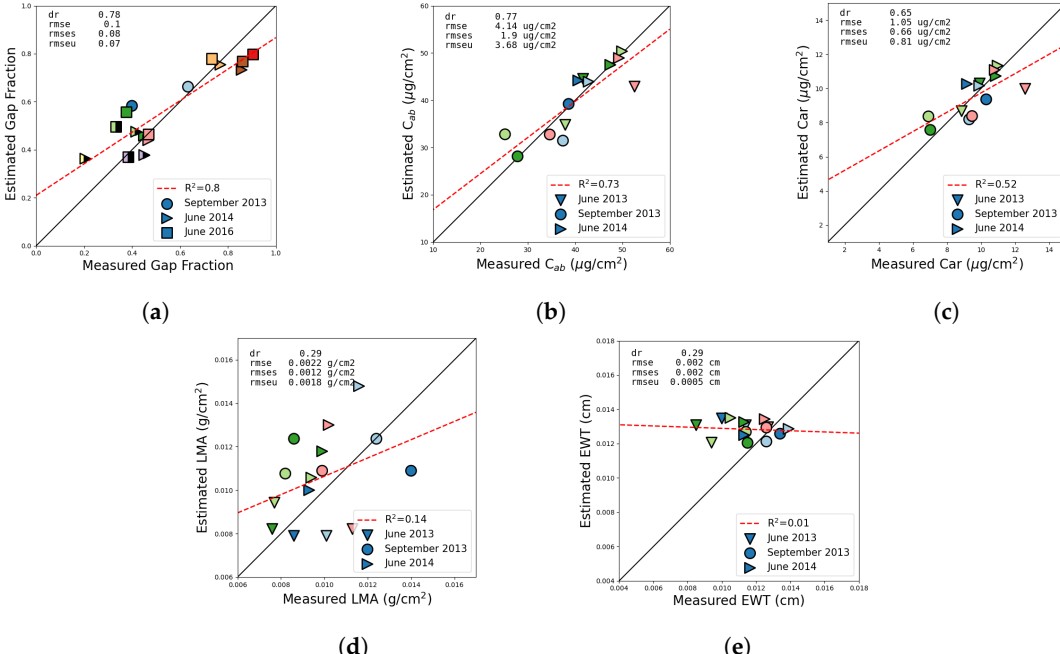

**Figure 9.** Comparison between estimated and measured parameters using the best-performing methods identified in Table 5. Panels (**b**–**e**) were obtained with the P2D fine LUTs, and panels (**a**,**d**) were obtained with the DART calibration LUTs. Marker color identifies the location within the study site, and for Gap Fraction QUDO-PISA mixed stands are identified with bicolored markers.

## 4. Discussion

### 4.1. PROSAIL2DART Performances

Choosing a sampling method for the variables' space for LUT generation, either for LUT-based inversions or calibration of machine-learning methods, is a critical step. No ideal sampling has yet be determined: Darvishzadeh et al. [55] and Hernández-Clemente et al. [22] used uniform distributions over the variables' intervals; Malenovsky et al. [56] and Leonenko et al. [57] considered a regular grid for LAI; and Richter et al. [58] and Verrelst et al. [59] used both uniform and Gaussian distribution for the parameters of interest. These methods can lead to different estimation accuracies, as they may put an emphasis on unadapted ranges or allow for the existence of duplicate or near-duplicate cases. However, as LUT-building is very time-consuming when using 3D RTMs, it is difficult to test different sampling methods.

As shown in Figure 5, the intrinsic error of the P2D approximation is very low for all wavelengths at CC 30% and gets lower when the CC increases, as the higher the CC the closer a DART scene is to a completely turbid medium. The error $E$ calculated in this study is also relative to the *cal* LUT's sampling, so lower absolute errors could be obtained by simply refining the calibration sampling.

This low intrinsic error made it possible to approximate with minimal error DART outputs using the PROSAIL model, which has considerably shorter computation times. Each interpolator necessitated a total of 784 (672 in the visible and 112 in the NIR and SWIR) DART reflectances. As specified in Section 3.3, the most extensive calibration LUT, encompassing 21,840 cases, necessitated 12,666 CPU hours to generate with DART 5.7.3v1078 and allowed for calculating 300,000 entries in 1.5 h, which is a significant decrease of the computation time. Even though the latest DART versions are significantly faster than the version used in this study, the execution time remains considerably longer than P2D as P2D is basically as fast as PROSAIL.

This short execution time makes it possible to test various sampling schemes and either use the output reflectances as is, as the P2D error is small, or determine an optimal sampling scheme to use for final LUT generation with the 3D RTM.

### 4.2. Gap Fraction Estimations

Table 5 and Figure 9a show that Gap Fraction estimations are good, with high $d_r$ and low RMSE, and that this is true no matter the estimation strategy. Indeed, all methods presented a $d_r$ higher than 0.7, a RMSE lower than 0.11, and a $R^2$ higher than 0.73 (not shown), and using the P2D LUTs did not lead to significant improvement of the estimations. The method presenting the highest $d_r$ was $D_{NDVI}$. When estimating scene LAI over the site, Miraglio et al. [30] also identified this method as the best performing. The LAI had been obtained from the same DHP pictures, with the assumption that effective plant area index (PAI) was equivalent to true LAI [60]. As effective PAI is derived from the DHP Gap Fraction, the fact that the same inversion strategy yielded good estimation results for both Gap Fraction and true LAI may confirm that true LAI can be considered equivalent to effective PAI for sparse broad-leaved forests.

### 4.3. Pigment Estimations

$C_{ab}$ and Car estimations (RMSE 4.14 and 1.85 µg/cm$^2$ and R$^2$ of 0.73 and 0.52, respectively) are in line with what can be found in the literature: Zarco-Tejada et al. [61] obtained a RMSE of 8.1 µg/cm$^2$ for $C_{ab}$ over open-canopy tree crops; le Maire et al. [43] obtained a RMSE of 8.2 µg/cm$^2$ estimating leaf $C_{ab}$ of broadleaved forests; Darvishzadeh et al. [62] had an RMSE of 8.6 µg/cm$^2$ when estimating leaf $C_{ab}$ of spruce stands; Zarco-Tejada et al. [63] obtained a 1.3 µg/cm$^2$ RMSE for Car estimation in vineyards with high-resolution imagery; Huang et al. [64] had a 2.02 µg/cm$^2$ RMSE when monitoring crop Car.

While final $C_{ab}$ estimation performances obtained in this study are similar or better than those obtained by Miraglio et al. [30], there are two main differences between the two methodologies: The first one is that the previous study used PROSPECT-5 and not PROSPECT-D: the principal differences between PROSPECT-D and -5 are the introduction of anthocyanins and the modification of the pigments specific absorption coefficients (SAC) at various wavelengths [34]: the chlorophylls SAC were lowered over the 0.45–0.65 µm interval and increased over the 0.65–0.7 µm, and the carotenoids SAC were increased over the 0.45–0.55 µm interval. The second difference is the sampling scheme, which was previously a regular grid over the variables' variation ranges, meaning that no correlation between $C_{ab}$ and Car, such as the one visible Figure 6, could be modeled. While this was not critical for $C_{ab}$ estimations, as the high $d_r$ values in Table 5 show, this limited the number of possible VI to use for Car estimations. Introducing the relationship between $C_{ab}$ and Car when building the LUTs allowed to use $C_{ab}$ VI for Car estimations, which proved to be more adequate. This relationship also helps to reduce the unnecessary cases in the LUTs, as low $C_{ab}$-high Car cases (and vice versa) are unrealistic and could bring confusion when doing the inversions.

*4.4. LMA and EWT Estimations*

The RMSE obtained for LMA was 0.0022 g/cm$^2$, more than two times higher than the $9.1 \times 10^{-4}$ g/cm$^2$ value obtained by le Maire et al. [43] over a higher LAI broadleaved forest. EWT estimations present a 0.002 cm RMSE. Neither LMA nor EWT present good $R^2$. As visible in Table 5, almost none of the VI tested for these variables showed a relationship with them. This is in line with the results found by Yanfang Xiao et al. [65], who showed using PROSAIL that for LAI lower than 3 it was not possible to estimate EWT as its contribution to the signal was too low in comparison to LAI's and ground's. It is possible that higher resolution hyperspectral images would be needed for EWT and LMA estimation, as this would make it possible to locate pure-vegetation pixels where EWT and LMA spectral signature would be more visible.

## 5. Conclusions

The results obtained in this study demonstrated the possibility to approximate with minimal error the reflectance outputs of DART with those of PROSAIL even at low (30%) CC. For higher CC, it was shown that approximation errors were negligible. The approximation model was further used to generate extensive LUTs to estimate Gap Fraction of mixed oak and pine stands as well as leaf C$_{ab}$, Car, EWT, and LMA of oak stands in a low-foliage woodland savanna. Gap Fraction and leaf pigment content estimations presented similar or improved performances when taking advantage of the proposed model instead of only relying on DART. EWT and LMA could not be retrieved using either models.

In summary, the findings show that acceptably approximating DART results from PROSAIL is possible and that the subsequent reflectances can be successfully used for estimation purposes of even very sparse oak stands, although conclusions should also applicable to other broadleaved stands due to the elementary modeling used in the 3D RTM. This is valuable, as 1D RTMs are dramatically faster than 3D RTMs. In the exploration phase, this allows for the testing of various sampling schemes at a negligible cost for either the training of machine learning methods, that require extensive training databases, or the generation of more complex LUTs. Approximated reflectances can also directly be used as is to retrieve canopy structural and biochemical parameters with acceptable accuracy.

Due to the tree distribution within the study site and the ground sampling distance of AVIRIS-C, no pine-dominant stands could be considered for Gap Fraction and leaf biochemistry estimations and this study focused mainly on pure-oak stands. Further work is necessary to extend them to coniferous trees or mixed stands. More work is also necessary to acceptably estimate EWT and LMA of tree–grass ecosystems, possibly by improving the soil realism by modeling the grass layer [66] or the tree representation with the inclusion of detailed trunk structures [16] within the 3D RTM.

**Author Contributions:** Conceptualization, T.M., S.U. and X.B.; Data curation, T.M., K.A. and M.H.; Formal analysis, T.M.; Funding acquisition, K.A., S.U. and X.B.; Investigation, T.M.; Methodology, T.M. and X.B.; Project administration, S.U. and X.B.; Resources, K.A., M.H. and S.U.; Software, T.M.; Supervision, S.U. and X.B.; Validation, T.M.; Visualization, T.M.; Writing—original draft, T.M. and M.H.; Writing—review and editing, K.A., M.H., S.U. and X.B. All authors have read and agreed to the published version of the manuscript.

**Funding:** This research was funded by the Office Nationale d'Études et de Recherches Aérospatiales (ONERA) and by Région Occitanie.

**Acknowledgments:** The authors are grateful to the CSTARS team for collecting and processing the field data, and to NASA JPL from providing AVIRIS-C data (NASA grant No. NNX12AP08G). They also thank Jean-Philippe Gastellu-Etchegorry from CESBIO for his insight and help concerning the DART simulations.

**Conflicts of Interest:** The authors declare no conflicts of interest. The funders had no role in the design of the study; in the collection, analyses, or interpretation of data; in the writing of the manuscript; or in the decision to publish the results.

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
