# Peer review of "Joint Use of PROSAIL and DART for Fast LUT Building: Application to Gap Fraction and Leaf Biochemistry Estimations over Sparse Oak Stands"

_remotesensing, doi:10.3390/rs12182925_

Round 1
Reviewer 1 Report
This study approximated the outputs of a 3D RTM from a 1D RTM and generated in very short time numerous extensive Look-Up Tables (LUT).The authors used the resultant LUT to estimate Gap Fraction, Cab, Car, EWT and LMA.
It is critical to precisely describe soil in the radiative transfer models for the study region of Tonzi Ranch.
Description of soil is bad in this study.
Methods, results and conclusions are not solid.
Author Response
Please see the attachment.
Reviewer 1
The authors used the resultant LUT to estimate Gap Fraction, Cab, Car, EWT and LMA.
It is critical to precisely describe soil in the radiative transfer models for the study region of Tonzi Ranch.
Description of soil is bad in this study.
Methods, results and conclusions are not solid.
The RTM soil in the present study was described as a lambertian surface. The reflectance of this lambertian surface was obtained by averaging the reflectances of multiple pixels of the AVIRIS-C images: these pixels were selected from open parts of the site. This is the same procedure as the one done in
Miraglio, T.; Adeline, K.; Huesca, M.; Ustin, S.; Briottet, X. Monitoring LAI, Chlorophylls, and Carotenoids Content of a Woodland Savanna Using Hyperspectral Imagery and 3D Radiative Transfer Modeling. Remote Sensing 2020, Vol. 12, Page 28 2019, 12, 28. doi:10.3390/RS12010028.
The above study found this simple modeling to be sucient to estimate LAI and leaf pigment content when working with AVIRIS-C images at an 18 m spatial resolution, which is the same spatial resolution as the present study. To account for possible spatial variation of the soil reflectance over the various plots, the present study also considered the mean std of the pixel reflectances when building the LUT. Precisions concerning the modeling within DART were added in Section 2.3.2.
We have enclosed a version of the revised manuscript with changes in blue for easier comparison with the initial submission.
List of changes
• Changes in the text are in blue
• Figure 1 and 2 were added to complement the description of the study site
• Figure 3 was added to complement section 2.2.1 and clarify the DHP acquisition method
• Figure 4 was updated to clarify the process of the approximation method
• Table 3 was updated to remove the PROSAIL column, that served little purpose
• Table 5 was updated with the new Gap Fraction values with the mixed plots
• Figure 7 and 9a were updated to include the added mixed Gap Fraction plots
1
Reviewer 2 Report
The proposed manuscript is a kind of report on a single study that has certain prospects and, more importantly, allows you to analyze certain difficulties arising in the studies of environmental and agricultural monitoring. The results contain specific conclusions and original approaches. The work can be published after revision.
Interesting results are presented in a somewhat confusing and misleading way. So in the description there are numerous details of the forest area. Meanwhile, the results of the work are concentrated in only one species of the described plants (blue oak), which extremely restricts the application of the proposed technology. It remains unclear how this relates to the state of other plants and whether this technology can be applied to other types of vegetation.
In this regard, the title of the article can hardly be retained, since the analysis of a plantation, and even more so its state for one species, cannot in any way reflect the data obtained in the work. I recommend that the authors change the title to reflect the actual results.
In addition, in the materials and methods, it is necessary to reflect the description of the object under study and bring its photo sheet, the size and shape of the sheet, the size and shape of the crown, other parts and their role in the modification of measurements. It is necessary to explain why the pines could not be involved in the analysis. It is not clear what the differences can be in the presence of damage to the crown by pests or phytopathogens, whether weather conditions affect the measurement parameters (temperature, humidity, precipitation ...). Probably the authors took these parameters into account, but they are not listed in this section. What is the proposed algorithm for accounting for such parameters?
Tables 1 and 2 need to be split as they are visually merged.
Figure 3. The signature does not reveal the method, if this is a scheme (which is likely), then it is not clear how objects with the same names (accompanied by arrows of different shapes) can be placed twice? Above the arrows, you should probably indicate the "action" or some algorithm of sequential actions, the method of obtaining data and the form of the report obtained as a result.
In table 3, the PROSAIL column contains no information, possibly not needed.
In table 3, the PROSAIL column contains no information, possibly not needed.
In table 4 LAI - the unit of measurement is sq.m. is it correct?
Biochemical measurements are important for this study, they fluctuate significantly depending on pressure, temperature and humidity, as well as other parameters such as pest and disease. Unfortunately, I do not find in the article a description of these parameters and methods of obtaining them, the number of repetitions. It's probably easy to fix by adding a description. Such an addition will make the article closer to specific rather than model problems, since I think it is extremely interesting to approximate the results from a practical perspective.
Figure 4 indicates the "summer" period, this is not accurate enough for such a study, since it is obvious that the leaves and metabolism of trees differ greatly by month, for this reason it is correct to compare within a specific month or close dates, it is better to indicate the month.
Additional illustrative material would make the article more useful and understandable.
The conclusion is very broad and does not reflect the concrete results and prospects. It seems to me correct to leave it as a discussion. Having formulated the main result obtained for oak in specific conditions and briefly the prospects for use on other objects and in other conditions.
I hope the comments will help improve the quality of your interesting article and the convenience for potential readers.
Author Response
Please see the attachment.
Reviewer 2
We thank reviewer 2 for their careful reading of the manuscript, comments and suggestions. We have enclosed with the answer to the their suggestions a version of the revised manuscript with changes in blue for easier comparison with the initial submission.
The proposed manuscript is a kind of report on a single study that has certain prospects and, more importantly, allows you to analyze certain difficulties arising in the studies of environmental and agricultural monitoring. The results contain specific conclusions and original approaches. The work can be published after revision. Interesting results are presented in a somewhat confusing and misleading way. So in the description there are numerous details of the forest area. Meanwhile, the results of the work are concentrated in only one species of the described plants (blue oak), which extremely restricts the application of the proposed technology. It remains unclear how this relates to the state of other plants and whether this technology can be applied to other types of vegetation.
In this regard, the title of the article can hardly be retained, since the analysis of a plantation, and even more so its state for one species, cannot in any way reflect the data obtained in the work. I recommend that the authors change the title to reflect the actual results.
In addition, in the materials and methods, it is necessary to reflect the description of the object under study and bring its photo sheet, the size and shape of the sheet, the size and shape of the crown, other parts and their role in the modification of measurements. It is necessary to explain why the pines could not be involved in the analysis. It is not clear what the differences can be in the presence of damage to the crown by pests or phytopathogens, whether weather conditions affect the measurement parameters (temperature, humidity, precipitation ...). Probably the authors took these parameters into account, but they are not listed in this section. What is the proposed algorithm for accounting for such parameters?
In order to address their first comment regarding the difficulties to apply the technology to other types of vegetation, additional Gap Fraction plots were included in the present study. These plots, while remaining oak-dominant, also include grey pines (up to 37% of the canopy cover). As shown in table 5 and Figure 9a, model performances remain similar, and estimation accuracy even for the most diverse stand remains in line with the one of pure-oak stands. The tree modeling done in the 3D RTM was very simple (lollipop representation), and blue oaks characteristics were only used to determine crown dimensions and canopy
height, so there is no reason to believe that conclusions should not be extendables to other broad-leaved trees (coniferous trees can require more complex modelling and would require further work, see Ref. 27 of the manuscript). Title, Section 2.1 and conclusion were updated to reflect this.
Concerning the second comment, aerial view of the site, with visible Gap Fraction plots and field measurement locations was added (Figure 1), as well as illustrations of a blue oak and a grey pine (Figure 2). Section 2.1 was updated to state more clearly that, as grey pines were not dominant in the AVIRIS-C
pixels (they only compose 10% of the overstory), it was decided not to model them in the 3D RTM.
Measurements of leaf chemistry to support interpretations of spectrally measured properties is very common in today's literature from authors in the U.S., Canada, Europe, China, Australia, etc. Web of Science identifies 55 papers with search \leaf chemistry and spectral measurements" and further searches
would have identified many more. Their data was collected following protocols similar to ours and we saw no evidence for environmental measurements of weather conditions and insects included in their results sections.
Below are a few examples:
• Covariance of Sun and Shade Leaf Traits Along a Tropical Forest Elevation Gradient By: Martin, Roberta E.; Asner, Gregory P.; Bentley, Lisa Patrick; et al. FRONTIERS IN PLANT SCIENCE Volume: 10 Article Number: 1810 Published: JAN 31 2020
• Limited Effects of Water Absorption on Reducing the Accuracy of Leaf Nitrogen Estimation By: Wang, Blowman J.; Chen, Jing M.; Ju, Weimin; et al. REMOTE SENSING Volume: 9 Issue: 3 Article Number: 291 Published: MAR 2017
• Leaf chlorophyll content retrieval from airborne hyperspectral remote sensing imagery By: Zhang, Yongqin; Chen, Jing M.; Miller, John R.; et al. REMOTE SENSING OF ENVIRONMENT Volume: 112 Issue: 7 Pages: 3234-3247 Published: JUL 15 2008
In particular, Greg Asner and Roberta Martens 2016 (Spectranomics: Emerging science and conser-vation opportunities at the interface of biodiversity and remote sensing By: Asner, Gregory P.; Martin, Roberta E. GLOBAL ECOLOGY AND CONSERVATION, Volume:8, Pages: 212-219, Published: OCT 2016) reviewed more than 10,000 leaf chemistry and leaf reflectance measurements from tropical regions, primarily South America, but their work includes Australia, Borneo, Puerto Rico Costa Rica, etc. (in sites with presumably variable atmospheric water vapor, insect pressures and other variables) and show very
detailed chemistry retrievals and their correlation with phylogenetic lineages in these forests.
Section 2.2.2 was updated to clearly state that collected leaves were healthy (showing no presence of pests or pathogens) and collected during dry days.
The introduction was updated in order to clarify the main goals of the study.
Tables 1 and 2 need to be split as they are visually merged.
Tables 1 and 2 were split.
Figure 3. The signature does not reveal the method, if this is a scheme (which is likely), then it is not clear how objects with the same names (accompanied by arrows of different shapes) can be placed twice? Above the arrows, you should probably indicate the "action" or some algorithm of sequential actions, the method of obtaining data and the form of the report obtained as a result.
Figure 6 was updated so that elements are only shown once. The various "
fluxes" between each action are now labelled to clarify the process.
In table 3, the PROSAIL column contains no information, possibly not needed.
The PROSAIL column was removed.
In table 4 LAI - the unit of measurement is sq.m. is it correct?
The unit of measurement for LAI is m2/m2. This represents the sum of the areas of the upper sides of the leaves in the scene, divided by the area of the scene.
Biochemical measurements are important for this study, they fluctuate significantly depending on pressure, temperature and humidity, as well as other parameters such as pest and disease. Unfortunately, I do not find in the article a description of these parameters and methods of obtaining them, the number of repetitions. It's probably easy to fix by adding a description. Such an addition will make the article closer to specific rather than model problems, since I think it is extremely interesting to approximate the results from a practical perspective.
See the above answer to the reviewer's comment. The present study focuses on major traits, such as total chlorophylls, total carotenoids, etc. Concerning remote sensing and plant traits, the factors that have the greatest impact on the spectral measurements are the light environment that the leaf developed in (e.g.,
sun/shade), the age of the leaf (for evergreen species that keep leaves for more than 1 year), how the leaves are handled after they are cut from the plant (usually people put them in tin foil and then into ziplock bags (or directly into ziplock bags), then into a cooler in the field so they are kept dark and cool. After that the length of time between harvesting them and measuring them is important, especially this is true for thin leaf agricultural species. In our case we are working on a species (blue oak and/or gray pine) that are extremely drought hardy and have adaptations that limit water losses.
Figure 4 indicates the "summer" period, this is not accurate enough for such a study, since it is obvious that the leaves and metabolism of trees differ greatly by month, for this reason it is correct to compare within a specific month or close dates, it is better to indicate the month.
All over the manuscript, "summer" was replaced by "June" and "fall" by "September" to properly identify the month.
Additional illustrative material would make the article more useful and understandable. The conclusion is very broad and does not reflect the concrete results and prospects. It seems to me correct to leave it as a discussion. Having formulated the main result obtained for oak in specific conditions and briefly the
prospects for use on other objects and in other conditions.
We thank the reviewer for his suggestion. Additional illustrations were added to better describe the study site and species (Figures 1 and 2) as well as the plots of interests for Gap Fraction (Figure 3). The conclusion was restructured to focus on the specfic results of the study, the possible applications of the proposed
approximation method, and to give additional prospects for futurs works.
I hope the comments will help improve the quality of your interesting article and the convenience for potential readers.
List of changes
• Changes in the text are in blue
• Figure 1 and 2 were added to complement the description of the study site
• Figure 3 was added to complement section 2.2.1 and clarify the DHP acquisition method
• Figure 4 was updated to clarify the process of the approximation method
• Table 3 was updated to remove the PROSAIL column, that served little purpose
• Table 5 was updated with the new Gap Fraction values with the mixed plots
• Figure 7 and 9a were updated to include the added mixed Gap Fraction plots
Reviewer 3 Report
Lines 84 and 85: What is the total land area (in hectares) of the study site?
Lines 98 and 99: It is unclear how the 60 x 60 m plots were chosen where the 9 DHPs were collected? Was a numbered grid (representing 60 x 60 m on the ground) digitally placed on an aerial image of the study area, then numbers from the grid randomly selected? If plots were not randomly selected then this could have biased the results.
Line 100: ‘Information concerning the number of plots for each date is given in Table 1’, yet the table caption reads ‘number of field points’. Which is it, a point or plot? In forestry or vegetation analyses, plots have some amount of land area that is being sampled (e.g. 1/10 hectare).
Line 101 and 102: How were the 9 locations of the 9 DHPs chosen in each plot? Were they laid out in cardinal directions or some other systematic way? Were the DSPs laid out identically in each plot?
Line 102: Tense change ‘ is visible’… why?
Lines 109 and 110: How were the five QUDO individuals chosen? Were they randomly selected? If they weren’t randomly selected then this could have biased the results.
Line 111: ‘to reach’ can be omitted.
Lines 114 and 115: Were leaves for LMA and EWT collected from the same five QUDO individuals where the Cab and Car leaves were taken? It is unclear from your description. Again, if the tree weren’t randomly selected this could have biased the results.
Lines 117 and 118: ‘Weighted’ should be changed to ‘weighed’. Definition of weighted from Merriam-Webster: made heavy, having a statistical weight attached, compiled or calculated from weighted data.
Lines 120 and 121: How long were the leaf samples dried at 65C? What kind of equipment was used to dry the leaves? How did you know they were all at a constant weight?
Line 22: Calculations for LMA and EWT should be placed before 2.2.2. Leaf Biochemistry.
Lines 124-126: How many trunks were measured? The same five from the leaf biochemistry collection? If not, how were the trunks chosen? What side of the tree were the reflectances measured?
Line 327: literature is misspelled
Author Response
Please see the attachment.
Reviewer 3
We thank reviewer 3 for their careful reading of the manuscript and improvement suggestions concerning the field data acquisitions. We have enclosed with the answer to the their suggestions a version of the revised manuscript with changes in blue for easier comparison with the initial submission.
Lines 84 and 85: What is the total land area (in hectares) of the study site?
The various Gap Fraction 60x60 m2 plots were obtained over an area spanning about 112 hectares (see Figure 1). An aerial view of the site was added with the site coordinates to show the dimensions of the site and repartition of the various plots.
Lines 98 and 99: It is unclear how the 60 x 60 m plots were chosen where the 9 DHPs were collected? Was a numbered grid (representing 60 x 60 m on the ground) digitally placed on an aerial image of the study area, then numbers from the grid randomly selected? If plots were not randomly selected then this could
have biased the results.
Several 60 x 60 meters plots were selected to span the full variation in species composition and canopy density. Each plot is divided into 9 subplots and 3 subplots were randomly selected for field measurements. Within each subplot selected, we sampled at three locations, forming a triangle approximately at the center of the subplot. The three sample points were located in a north, southeast and southwest orientation (see Figure 3). Section 2.2.1 was updated to clarify this.
Line 100: `Information concerning the number of plots for each date is given in Table 1', yet the table caption reads `number of field points'. Which is it, a point or plot? In forestry or vegetation analyses, plots have some amount of land area that is being sampled (e.g. 1/10 hectare).
"point" was replaced by "data". It is now specified that it is indeed Gap Fraction plots that are being considered. The area of the plot is, as specified in Section 2.2.1, 3600 m2 or 0.36 hectare.
Line 101 and 102: How were the 9 locations of the 9 DHPs chosen in each plot? Were they laid out in cardinal directions or some other systematic way? Were the DSPs laid out identically in each plot?
See answer above about Lines 98 and 99.
Line 102: Tense change ` is visible'. . . why?
The tense of the sentence was modified to "was visible".
Lines 109 and 110: How were the five QUDO individuals chosen? Were they randomly selected? If they weren't randomly selected then this could have biased the results.
The criteria for selecting QUDO individuals were that 1- the tree canopy was not shaded by surrounding trees; 2- trees were "healthy"; 3- the trees looked typical of the site. Section 2.2.2 was updated to detail these criteria.
Line 111: `to reach' can be omitted.
"to reach" was removed.
Lines 114 and 115: Were leaves for LMA and EWT collected from the same five QUDO individuals where the Cab and Car leaves were taken? It is unclear from your description. Again, if the tree weren't randomly selected this could have biased the results.
The leaves were collected from the same QUDO individuals. Section 2.2.2 was updated to clarify this.
Lines 117 and 118: `Weighted' should be changed to `weighed'. Denition of weighted from Merriam-Webster: made heavy, having a statistical weight attached, compiled or calculated from weighted data.
Correction done.
Lines 120 and 121: How long were the leaf samples dried at 65C? What kind of equipment was used to dry the leaves? How did you know they were all at a constant weight?
The leaf samples were dried in a drying oven until constant weight was measured (around 2 to 3 days). They were then weighed, and reweighed later the same or next day. Samples were kept in the drying oven until no weight change was measured. Section 2.2.2 was updated to clarify this.
Line 22: Calculations for LMA and EWT should be placed before 2.2.2. Leaf Biochemistry.
Equations were moved to be before section 2.2.2.
Lines 124-126: How many trunks were measured? The same ve from the leaf biochemistry collection? If not, how were the trunks chosen? What side of the tree were the reflectances measured?
Trunk reflectances were measured on the leaf biochemistry trees. Small portions of the trunk were collected and situated in a horizontal surface to facilitate the measurement. Section 2.2.3 was updated to clarify this.
Line 327: literature is misspelled
Correction done.
List of changes
• Changes in the text are in blue
• Figure 1 and 2 were added to complement the description of the study site
• Figure 3 was added to complement section 2.2.1 and clarify the DHP acquisition method
• Figure 4 was updated to clarify the process of the approximation method
• Table 3 was updated to remove the PROSAIL column, that served little purpose
• Table 5 was updated with the new Gap Fraction values with the mixed plots
• Figure 7 and 9a were updated to include the added mixed Gap Fraction plots
Round 2
Reviewer 2 Report
The corrections made the article more understandable and I hope it will be highly appreciated by the readers of the journal, and the work will find further development.
In my opinion, the information obtained in this work is very interesting, innovative, and arise knowledge with high importance for the understanding forest development and study of growing.
So, I believe that this paper is within the quality parameters of the Remote Sensing Journal.